# Cardiac Toxicity Induced by Long-Term Environmental Levels of MC-LR Exposure in Mice

**DOI:** 10.3390/toxins15070427

**Published:** 2023-06-30

**Authors:** Canqun Yan, Ying Liu, Yue Yang, Isaac Yaw Massey, Linghui Cao, Muwaffak Al Osman, Fei Yang

**Affiliations:** 1Department of Health Management Center, The Second Affiliated Hospital, Hengyang Medical School, University of South China, Hengyang 421009, China; 1992020003@usc.edu.cn; 2Hunan Province Key Laboratory of Typical Environmental Pollution and Health Hazards, School of Public Health, Hengyang Medical School, University of South China, Hengyang 421009, China; liuying@usc.edu.cn (Y.L.); mriymassey@csu.edu.cn (I.Y.M.); 3Hunan Provincial Key Laboratory of Clinical Epidemiology, Xiangya School of Public Health, Central South University, Changsha 410017, China; yangy930806@126.com (Y.Y.); 206911027@csu.edu.cn (M.A.O.); 4Changsha Central Hospital, Changsha 410004, China; clh223234@126.com; 5Laboratory of Ecological Environment and Critical Human Diseases Prevention of Hunan Province, School of Basic Medical Sciences, Hengyang Medical School, University of South China, Hengyang 421009, China

**Keywords:** microcystins, chronic toxicity, myocardial fibrosis, PI3K/AKT/mTOR signaling pathway

## Abstract

Cyanobacterial blooms are considered a serious global environmental problem. Recent studies provided evidence for a positive association between exposure to microcystin-LR (MC-LR) and cardiotoxicity, posing a threat to human cardiovascular health. However, there are few studies on the cardiotoxic effects and mechanisms of long-term low-dose MC-LR exposure. Therefore, this study explored the long-term toxic effects and toxic mechanisms of MC-LR on the heart and provided evidence for the induction of cardiovascular disease by MC-LR. C57BL/6 mice were exposed to 0, 1, 30, 60, 90, and 120 μg/L MC-LR via drinking water for 9 months and subsequently necropsied to examine the hearts for microstructural changes using H&E and Masson staining. The results demonstrated fibrotic changes, and qPCR and Western blots showed a significant up-regulation of the markers of myocardial fibrosis, including TGF-β1, α-SMA, COL1, and MMP9. Through the screening of signaling pathways, it was found the expression of PI3K/AKT/mTOR signaling pathway proteins was up-regulated. These data first suggested MC-LR may induce myocardial fibrosis by activating the PI3K/AKT/mTOR signaling pathway. This study explored the toxicity of microcystins to the heart and preliminarily explored the toxic mechanisms of long-term toxicity for the first time, providing a theoretical reference for preventing cardiovascular diseases caused by MC-LR.

## 1. Introduction

Globally, increased eutrophication of water bodies and the proliferation of cyanobacteria have become a major public health issue. Cyanobacterial blooms are typically characterized by the massive formation of cyanobacteria and diatoms, which resulted in hypoxia and a bad smell. In addition, cyanobacteria such as *Microcystis*, *Anabaena*, *Arabidopsis*, *Oscillatoria,* and *Nostoc* [1] produce various potent toxins such as cyclic peptides, alkaloids, and lipopolysaccharides during their metabolism [2]. Exposure to these toxins has negative effects on human and animal health [3]. Among the cyanobacterial toxins produced during outbreaks of cyanobacterial blooms, monocyclic peptide MCs are the most abundant, widely distributed, highly toxic, and difficult to remove after being dissolved in water [4,5,6]. Therefore, the consequences of MCs on the environment, humans, and animals have attracted much attention in research. At least 279 MC variants have been documented, with MC-LR being the most toxic [7,8]. Human exposure to MCs can occur through various routes, including ingestion of ‘contaminated’ water during hemodialysis, consumption of drinking water, physical contact with the water, and engaging in recreational activities in waterbodies during algal blooms. However, drinking water remains the main route of exposure [3]. Studies have shown that MC-LR plays a critical role in damaging organ cells under both acute and chronic exposure. The liver [9,10], kidney [11,12,13], nervous system [14], gastrointestinal tract [15,16], reproductive system [17,18,19], and cardiovascular system [20,21] have been reported to be the target organs for MC toxicity. To minimize the health hazards caused by MCs, the World Health Organization (WHO) recommended that the maximum allowable content of MCs in drinking water should not exceed 1 μg/L [22].

Previous research has shown MCs can be transported into various cells by the organic anion-transporting polypeptide (OATP) [23]. Several OATP family genes, including OATP4A1, OATP2A1, OATP2B1, and OATP3A1, have been reported to be expressed in cardiac tissue [24]. This infers MCs can be transported to and accumulated in cardiac tissue cells, ultimately endangering the cardiac tissue. Zhao et al. [25] found that MC-LR directly induces cardiovascular toxicity by altering the cardiomyocyte morphology, status of apoptosis and proliferation, cytoskeleton, rhythm, differential expression/activity of transcription factors, ultrastructure of cardiomyocyte mitochondrial respiratory chain, and metalloproteinase activity when rabbits were injected with 12.5 and 50 μg/kg MC-LR (equal 75 and 300 μg/L in mice [26]) for 48 h. MC-LR exposure also enhanced the production of reactive oxygen species (ROS) and endoplasmic reticulum oxidative stress, leading to cytoskeleton disruption, as well as the dysfunction of mitochondria and the endoplasmic reticulum [27]. Milutinovic et al. [28] found that MC-LR could induce the enlargement of cardiomyocytes, loss of cell cross-striations, lower myofibril volume fraction, fibrosis, and mononuclear infiltration in the interstitial tissue in male adult Wistar rats, which were gavaged every second day for 8 months with MC-LR (10 μg/kg in rats equals 1687.5 μg/L in mice). Previous studies focused on the cardiovascular toxicity of short-term high-dose MC-LR, and no studies have investigated the toxicity and mechanisms of long-term low-dose exposure to MC-LR through drinking water and its effects on the heart [29].

To date, information regarding the cardiotoxic effects of MC-LR, especially low concentration of long-term MC-LR exposure, remains limited, and the underlying mechanism is still unclear. The majority of previous investigations examined the acute effects of exposure to high-concentration MC-LR on the heart. This study is designed to explore the effects of low concentration (1, 30, 60, 90, and 120 μg/L MC-LR) MC-LR (WHO guideline 1 μg/L) following chronic exposure on the heart and to elucidate the potential mechanism of MC-LR cardiotoxicity including the involvement of the PI3K/AKT pathway according to in vivo experiments. The goals of this study are to further understand the mechanism of microcystin cardiotoxicity and provide a theoretical basis and new perspective on the prevention, diagnosis, and treatment of microcystin-induced cardiotoxicity.

## 2. Results

### 2.1. Detection of Endogenous Exposure to MC-LR in Cardiac Tissue

To detect the presence of internal exposure of the heart tissue to MC-LR, total heart protein was extracted, and a Western blot was performed. Except for the control group, MC-LR was detected in all MC-LR treatment groups (Figure 1). This suggests MC-LR entered the heart tissue via absorption in mice. The concentration of MC-LR increased in the heart when the oral MC-LR dose increased.

### 2.2. The Effect of MC-LR on the General Condition and the Weight Index of the Mouse Heart

During the period of exposure, the mice’s body weights were recorded every two weeks (Figure 2A). No deaths or abnormal behaviors were observed in the mice during the course of the experiment. Compared with the control group, there were no significant differences in body weight change compared to controls over the nine-month course of this study. The volume of drinking water consumed by the mice was measured each week during the exposure period (Figure 2B). In fact, the results of Figure 2A showed that the total amount of water drunk by five mice per cage was about 300 mL per month, which was equivalent to drinking 10 mL of water per day, that was, each mouse drank 2 mL of water per day. There was no significant change in the amount of drinking water consumed among the MC-LR exposure groups. The data of the drinking water and exposure doses of mice and the average daily intake of MC-LR for each mouse in each dose group were estimated (Figure 2C). The heart weights for each mouse are normalized to their body weight. The mouse heart weight index is shown that there were no significant changes in the cardiac weight index compared with the control group (Figure 2D).

### 2.3. Effects of MC-LR Treatment on Mouse Cardiac Function

The results demonstrated that the heart function of mice in the highest exposure (120 µg/L) group decreased significantly. There was an absence of differences in the other treatment groups (Table 1).

### 2.4. Effects of MC-LR Treatment on the Histological Morphology of Mouse Heart

The mice’s exposure to MC-LR in drinking water for 9 months resulted in observed changes in cardiac histology. Cardiac micromorphology is shown in Figure 3A,B, indicating the presence of myocardial fibrosis. There was an absence of differences in the other treatment groups. To further clarify the phenotype of myocardial fibrosis, Masson staining was performed to observe the abundance of collagen-specific tissues. An increase in blue collagen fibers can be observed as a diffuse pattern of enhanced collagen interspersed between myocardial fiber bundles. There was an absence of differences in the other treatment groups (Figure 3C,D).

### 2.5. Effects of MC-LR on the Expression of Myocardial Fibrosis Marker Factors in Mouse Heart Tissue

To further clarify the myocardial fibrosis caused by MC-LR, qPCR was used to determine the presence and abundance of mRNA levels for myocardial fibrosis marker factors TGF-β1, α-SMA, and Col1. The relative levels of mRNA levels for TGF-β1, α-SMA, and COL1 mRNA were significantly increased in all MC-LR exposure groups compared to the control group (Figure 4).

In addition, changes in protein levels of these myocardial fibrosis marker factors were determined via Western blot. The results are shown in Figure 5. Compared to the control group, the protein levels of all measured cardiac fibrosis markers were significantly up-regulated at all doses of MC-LR treatment.

### 2.6. MC-LR Expression of PI3K/AKT/mTOR Pathway-Related Proteins in the Mouse Heart

To more fully elucidate the mechanism of MC-LR-induced cardiotoxicity, Western blotting was used to detect the expression levels of p-PI3K, PI3K, p-AKT, AKT, p-mTOR, and mTOR protein. Compared to the control, the relative expression of p-PI3K/PI3K protein did not change significantly in the 1 μg/L MC-LR exposure group but was significantly elevated in the 30, 60, 90, and 120 μg/L MC-LR exposure groups (*p* < 0.05) to confirm MC-LR can activate the PI3K/AKT/mTOR pathway. Similarly, compared with the control group, the relative expression of p-mTOR/mTOR protein did not significantly change in the 1 μg/L MC-LR exposure group but was significantly elevated in the 30, 60, 90, and 120 μg/L MC-LR exposure group. The relative expression levels were all significantly increased (*p* < 0.05), as shown in Figure 6.

## 3. Discussion

This present study investigated the effects of MC-LR on the heart using a chronic oral exposure regimen administered in a dosimetric fashion. The heart tissues were examined for internal exposure dosing after 9 months of oral MC-LR exposure. As anticipated, except for the control group, algal toxins were detected in all MC-LR-treated groups. H&E and Masson’s staining were applied to observe the microstructure of the heart, while changes in mRNA expression levels and protein levels in myocardial fibrosis-related factors were measured.

Mice demonstrated 100% survival following 9 months of MC-LR treatment. No significant differences in body weight gain, diet, water consumption, or activity levels were observed. No abnormality in the heart weight ratio between the MC-LR-treated group and the control group was noted. Du et al. [30] treated mice with low-dose MC-LR for one month without significant changes in body weight; these results are consistent with our results. The findings of this study showed that chronic exposure to MC-LR could decrease heart function in mice, and this study is likely the first or among the first to demonstrate such a decrease. To date, there is no research showing MC-LR leads to a decline in heart function. Previous research shows MC-LR suppressed heart rate. For example, Qiu et al. [31] injected MC-LR with high concentrations of 0.16 LD_50_ (14 g/kg) and 1LD_50_ (87 g/kg), and there is a drop in heart rate and blood pressure in survival rats [31]. Martins et al. [32] injected 100 μg/kg MC-LR into *Hoplias malabaricus* induced cytotoxicity by enhancing the activity of oxidative stress and other related enzymes. Saraf et al. [33] found that MC-LR can suppress the heart rate of *Oryzias Latipes*. Qi et al. [34] found that zebrafish (*Danio rerio*) larvae had morphological deformations, growth retardation, heart rate inhibition, and increased apoptosis after 96 h of treatment with 4.0 mM MC-LR. In contrast, the results of this study found the heart rate of the mice in all MC-LR treatment groups was not significantly different from that of the control group (Table 1). Interestingly, this present research first found MC-LR changed heart function, including LVEF, LVFS, LVESD, and LVEDD. These differences in the findings suggested that there may be some differences in the toxic mechanism of acute and chronic exposure to MC-LR, as well as differences in concentrations of MC-LR. In the future, the heart rate of the other lower treatment group (1, 30, 60, and 90 µg/L) will be further performed to reveal the heart function in mice.

Milutinovic et al. [28] found that the appearance and morphology of the heart of animals treated with MC-LR did not change significantly for eight months. TUNEL staining showed no change in apoptosis, but the cytoskeleton structure of cardiomyocytes was destroyed, with myocyte striations absent and myofibril volume fraction reduced. Further observations were an increase in cardiomyocyte volume, a decrease in the volume of myofibrils, and the presence of fibrosis, along with monocyte infiltration in the interstitial tissues of the heart [28]. Suput et al. [35] used MC-LR to conduct a similar chronic toxicity study and demonstrated a reduction in the volume and density of the myocardium, accompanied by fibrous proliferation and limited infiltration of lymphocytes. These findings indicated MC-LR can induce myocardial fibrosis, which is consistent with ours.

Cardiac fibrosis is a common pathological change in many advanced cardiovascular diseases, including ischemic heart disease, hypertension, and heart failure. Myocardial fibrosis can affect cardiac function by increasing myocardial stiffness and impairing conduction, which have been identified as common risk factors for heart failure and arrhythmias [36]. Cells in the heart express profibrotic factors, cytokines, and chemokines, such as α-SMA, TGF-β1, and COL1 after stimulation by profibrotic factor signals and cardiac injury. These factors bind to their corresponding receptors and activate signaling pathways that regulate cardiac fibrosis [37]. Furthermore, once profibrotic growth factors are secreted in fibroblasts of the heart, these factors form a positive feedback regulation, which, in turn, amplifies the fibrotic signal and ultimately promotes the development of cardiac fibrosis [25]. Fibroblasts make up the majority of mammalian heart cells, and they are primarily responsible for the production of major components of the extracellular matrix, including COL (I, III, V, and VI) and fibronectin. MMP-9 can be used as a marker for serious changes in the heart and would be associated with inflammation and fibrosis, which was over-expression in this present study [38]. The occurrence of myocardial fibrosis can pose a serious threat to human health. However, information regarding the cardiotoxic effects of MC-LR, especially low concentration of long-term MC-LR exposure, remains limited, and the underlying mechanism is still unclear. This study first demonstrated that long-term low concentrations of MC-LR (120 μg/L) exposure can induce myocardial fibrosis in mice.

The PI3K/AKT signal pathway is a family of key enzymes involved in various biological processes, including cell survival, apoptosis, metastasis, differentiation, metabolism, protein synthesis, cell polarity, and motility, as well as vesicle transport and cardiac function [39]. Numerous studies have shown that PI3K is involved in the process of cardiac fibrosis. Data from a previous study [40] demonstrated that enhancing the PI3Kα signal modulates, including TGF-β, and miR-21 expression, further protected cardiac fibrosis in pathological hypertrophy [41]. Ma et al. [42] found that overexpression of AKT reduces piperine-mediated protection of cardiac fibroblasts. Inhibition of AKT significantly reduces pressure-load-induced cardiac fibrosis [43]. mTOR is a direct substrate of AKT kinase. Finckenberg et al. [44] found that when the mTOR pathway was blocked, the expression levels of pro-fibrotic factors such as connective tissue growth factor, TGF-β1, and Col3 were significantly down-regulated. These studies suggested that the PI3K/AKT/mTOR signal pathway may be involved in the regulation of cardiac fibrosis.

PI3K may regulate fibrosis by affecting myocardial contractility [45]. AKT, also known as protein kinase B (PKB), is a serine ser/threonine protein kinase that regulates cellular functions, including survival, growth, and metabolism [46]. Over-expression of AKT1 and AKT3 resulted in increased cardiomyocyte size and enhanced function [47]. Mitra et al. [48] found that OSI-027, an mTOR inhibitor, inhibited the expression of TGF-β-induced myocardial fibrosis marker factors α-SMA, Col I and Col III. The study of Liu et al. showed the PI3K/AKT signaling pathway was activated in the HL7702 cell line under the induction of MC-LR but not in a dose or time-dependent fashion [49]. Chen et al. also found the PI3K/AKT signaling pathway was activated in testes but not in a dose-time-dependent manner [50]. In addition, the proteomics data from Zhou [51] also found that the PI3K/AKT signaling pathway was activated in the testis. The results of these studies are consistent with this current study’s findings.

To further study the mechanism of MC-LR-induced myocardial fibrosis in mice, the signaling pathway was examined. It was found that MC-LR can up-regulate the expression of phosphorylated PI3K, AKT, and mTOR to activate the PI3K/AKT signaling pathway (Figure 7). Phosphorylated PI3K and phosphorylated mTOR were significantly up-regulated following 9 months of treatment at 30, 60, 90, and 120 μg/L of MC-LR. There was no significant change in the 1 μg/L dose group. Phosphorylated AKT was significantly up-regulated in all MC-LR treatment groups. However, the change in protein levels of these signaling molecules was not dose dependent. The potential reason is the low dose (1 μg/L) of MC-LR is insufficient to cause changes in protein levels, or the changes caused do not attain the threshold of detection. At higher concentrations (30 μg/L, 60 μg/L, 90 μg/L, and 120 μg/L), MC-LR may trigger protection mechanisms to elicit change. The mutual antagonism of toxic effects resulted in the alteration of the amount of protein, although not in a dose–response relationship. The specific mechanism remains to be further studied and elucidated.

## 4. Conclusions

In summary, we first demonstrated that MC-LR may induce myocardial fibrosis by activating the PI3K/AKT/mTOR signaling pathway. This study explored the toxicity of microcystins to the heart and preliminarily explored the toxic mechanisms of long-term toxicity for the first time, which provides a theoretical reference for the prevention of cardiovascular diseases caused by MC-LR.

## 5. Materials and Methods

### 5.1. Materials

#### Grouping and Handling of Experimental Mice

Sixty specific pathogen-free (SPF) C57BL/6 male mice, 6–8 weeks of age and weighing 18–22 g, were purchased from the Hunan Slack Laboratory Animal Company and raised in the Animal Center of the Xiangya School of Public Health at Central South University. Animal housing conditions were maintained at an ambient temperature of 21–22 °C, humidity of 75 ± 5%, and kept on a 12 h light/dark cycle. Clean rodent pellets and drinking water were provided to allow mice to eat and drink freely. Following two weeks of adaptive feeding, the mice were divided into 6 groups, based on the method of random number generation, with each group consisting of 10 mice. For each group, mice were housed in 2 cages, 5 mice per cage. MC-LR (purity ≥ 95%) was obtained from Alexis Corporation (Lausen, Switzerland) and stored at −20 °C. The MC-LR is first dissolved in DMSO and then diluted into water. Animals were provided drinking water with 0, 1, 30, 60, 90, or 120 μg/L MC-LR. The drinking water with MC-LR was changed every week. The MC-LR concentrations of drinking water were measured before and after feeding by HPLC and were not changed. Documentation was made weekly of the volume of drinking water consumed. Food was also provided promptly, adequately supplied, and freely available. Mouse body weight was measured every two weeks. The exposure time was for a total time of 9 months [30]. All animal studies were approved by the Ethics Research Committee of the Xiangya School of Public Health, Central South University (approval number, XYGW-2018-41, and date of approval: 10 November 2018).

### 5.2. Experimental Method

#### 5.2.1. H&E Staining

The C57BL/6 mouse hearts were quickly removed, washed with pre-cooled saline, and immediately fixed in 4% paraformaldehyde overnight. Hearts (1/3 of the heart) were subsequently embedded in paraffin and cut into 4μm thick sections. Tissue sections were dewaxed using xylene and ethanol, washed with distilled water, and stained with hematoxylin stain to visualize the cell nucleus and intranuclear proteins for 5–15 min. Excess dye was washed from each slide with distilled water. Additionally, 1% hydrochloric acid was used for 10 s to further stain the nucleus. All slides were rinsed with running tap water for 15–30 min. A saturated solution of lithium carbonate was used to form a temporary alkalization condition to enhance the blue staining of the nucleus for all cells. After washing with distilled water, the heart sample was stained with eosin dye solution for 1–5 min and then placed in different concentrations of alcohol for dehydration. After drying at room temperature, all sections were coverslipped with a neutral mounting medium resin. Morphological changes were observed under a microscope, and images were obtained and analyzed. Each test group consisted of heart tissues from 5 mice of the different treatment groups.

#### 5.2.2. Masson Staining

Paraffin sections were placed in xylene for 15 min for deparaffinization, followed by a graded series of ethanol (100, 90, 85%) and final rinsing with distilled water for 2 min. Tissue sections were placed in boudin solution at 56 °C for 1 h, rinsed with running water, followed by placement in lapis lazuli blue dye solution for 5 min, hematoxylin staining for 5 min, rinsed with water, treated with 1% hydrochloric acid alcohol, followed by rinsing with water. Tissue sections were subsequently placed in ponceau acid fuchsin dye solution, rinsed, and serially treated with 1% phosphomolybdic acid, bright green, 1% glacial acetic acid, and a graded series of ethanol (95, 100%) and xylenes, before coverslipping with neutral resin mounting media. This procedure stains collagen fibers blue and muscle fibers red.

#### 5.2.3. Echocardiography

Echocardiography was performed using a Vevo 2100 ultra-high resolution small animal ultrasound imaging system and MS-550 probe. Left ventricular long-axis view, parasternal left ventricular long-axis view, suprasternal view, and apical four-chamber view were performed. At the parasternal left ventricular long-axis view, M-mode ultrasound was used to record the motion curve of the left ventricular wall. The measured parameters included left ventricular end-systolic diameter (LVESD), left ventricular end-diastolic diameter (LVEDD), left ventricular end-diastolic posterior wall (LVPWd), LVPWs (left ventricular end-systolic posterior wall), heart rate (HR), left ventricular end-systolic stroke volume (ESV), left ventricular end-systolic stroke volume, left ventricular end-diastolic stroke volume (EDV), and left ventricular end-diastolic stroke volume. The left ventricular short-axis shortening rate was calculated (left ventricular fraction shortening, FS %) using the following formula: FS% = (LVEDD − LVESD)/LVEDD × 100%. The left ventricular ejection fraction (LVEF%, left ventricular ejection fraction) was also calculated using the following formula: LVEF% = SV (stroke volume)/EDV × 100%. Cardiac output CO (cardiac output) was determined using the following formula: CO = ESV × HR. Results were recorded and analyzed three times.

#### 5.2.4. qPCR

The heart was quickly removed from the thorax, placed on ice, and washed with pre-cooled PBS. RNA was extracted using the Trizol method following tissue homogenization. The extracted RNA was reverse transcribed to synthesize cDNA according to the instructions of the Weizan reverse transcription kit following tissue homogenization. The qPCR reaction was performed according to the steps of the Weizan fluorescence quantitative kit instructions (Table 2). The reaction conditions were 95°, 30 s pre-denaturation; followed by 95°, 10 s denaturation; and 60°, 30 s annealing/extension for a total of 40 cycles. After the cycle, the temperature was raised to 95 °C for the dissolution curve. The 2^−∆∆Ct^ comparison method was used to calculate the transcription fold change of target genes in each treatment group relative to the average value for the control group. Fold-change calculations were based on the following equations: △Ct = Ct (target) − Ct (internal reference), △△Ct = △Ct (MC-LR exposure group) − △Ct (control group). GraphPad Prism software was used to create statistical graphs.

#### 5.2.5. Western Blot

At necropsy, the heart was quickly removed, placed on ice, washed with cold PBS, and dissolved in the RIPA lysis buffer (1 mL with 10 μL protease inhibitor) at 100 mg/mL. The sample was ground in a homogenizer for 180 s at 70 hertz (HZ) and centrifuged at 12,000 rpm (4 °C) for 15 min to prepare the tissue supernatant. Total protein concentration was determined according to the BCA method. Proteins were separated by 10% SDS-PAGE gel, transferred to Polyvinylidene Fluoride (PVDF) membrane, and blocked with 5% BSA and non-fat dry milk for 1 h at room temperature. The PVDF membrane was placed in the corresponding primary antibody incubation solution prepared in advance and incubated at 4 °C overnight (Table 3). A secondary antibody solution was applied for 1 h. ECL developer solution and a chemiluminescence imager were used, and the gray value of protein bands was analyzed using Image J software.

#### 5.2.6. Statistical Analysis

SPSS 25 software was used for the statistical analysis of data. Two independent sample *t*-tests were used for the comparison of two groups of samples. One-way analysis of variance was used for comparison among multiple groups of samples, and the LSD test was used for further pairwise comparisons. The significance of the two-sided test level was set at α = 0.05.

## Figures and Tables

**Figure 1 toxins-15-00427-f001:**
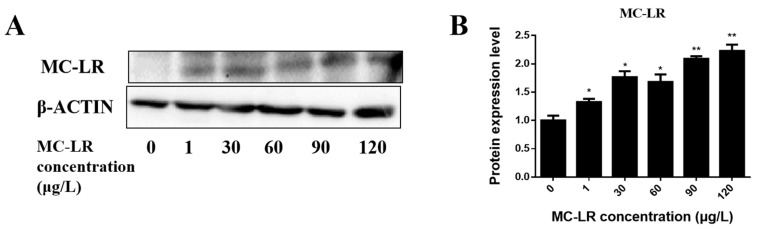
Detection of MC-LR in cardiac tissue. (**A**) Representative bands of Western blotting. (**B**) showed the quantitative results of MC-LR via Western blot, respectively. (* indicates a significant difference at *p* < 0.05, and ** indicates a significant difference at *p* < 0.01; *n* = 3).

**Figure 2 toxins-15-00427-f002:**
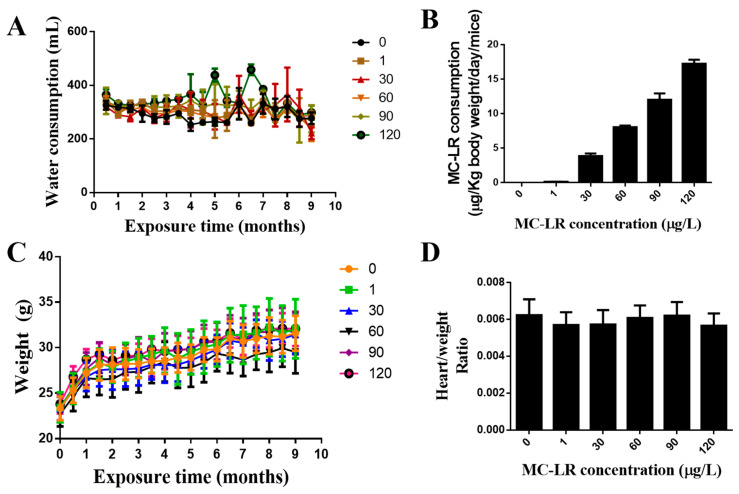
Changes in the consumption of water (**A**), the intake of MC-LR in every dose group (**B**), the body weight (**C**), and the mouse heart weight index (**D**), by the C57BL/6 mice during a 9-month chronic exposure to different concentrations of MC-LR (*n* = 10).

**Figure 3 toxins-15-00427-f003:**
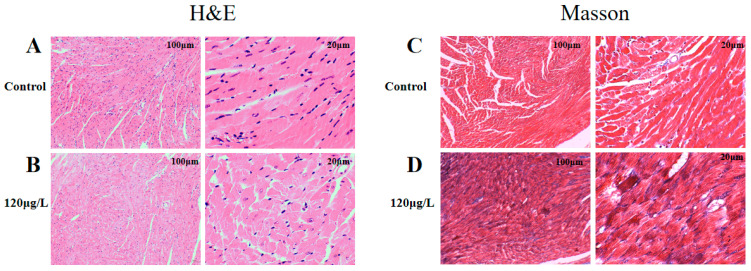
The effect of MC-LR treatment on the morphology of mouse nodal heart. H&E staining (**A**) represents the Control group; (**B**) represents the 120 µg/L MC-LR exposure group (*n* = 3). Masson Staining (**C**) represents the Control group; (**D**) represents the 120 µg/L MC-LR exposure group (*n* = 3). (Blue represented collagen fibers, and red represented muscle fibers and erythrocytes).

**Figure 4 toxins-15-00427-f004:**
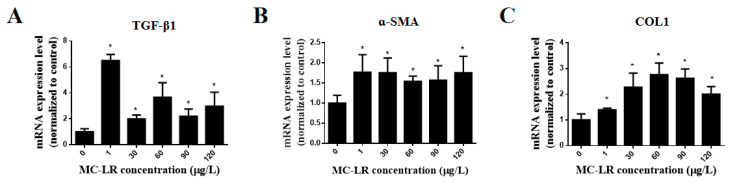
The expression of myocardial fibrosis factors TGF-β1 (**A**), α-SMA (**B**), and COL 1 (**C**) by the C57BL/6 mice during a 9-month chronic exposure to different concentrations of MC-LR. (* indicate significant difference at *p* < 0.05, *n* = 3).

**Figure 5 toxins-15-00427-f005:**
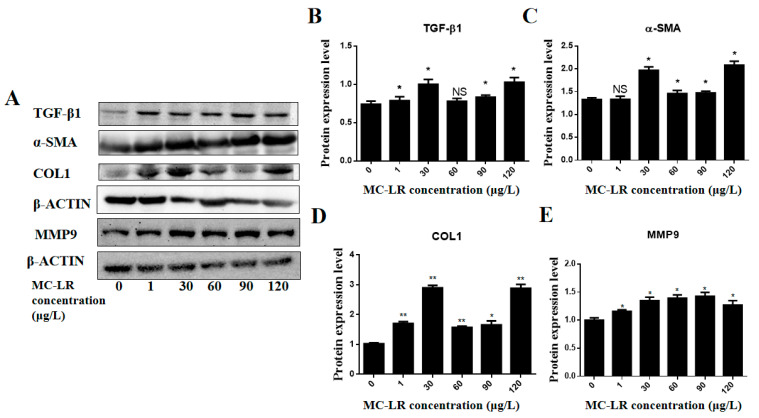
Western blot detection of the effect of MC-LR on the expression of myocardial fibrosis factors in mouse heart tissue. (**A**) represents Western blot results, and (**B**–**E**) showed the quantitative results of TGF-β1, α-SMA, COL1, and MMP9 via Western blot, respectively. (NS indicates a significant difference at *p* > 0.05, * indicates a significant difference at *p* < 0.05, and ** indicates a significant difference at *p* < 0.01; *n* = 3).

**Figure 6 toxins-15-00427-f006:**
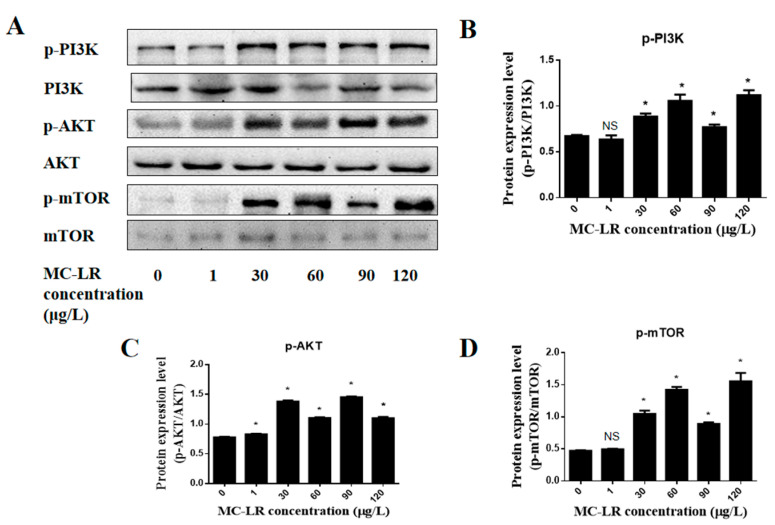
Effects of MC-LR on PI3K/AKT/mTOR pathway-related proteins in mouse heart tissue. (**A**) Related protein bands detected via Western Blot; (**B**–**D**) Statistical bar graph of protein expression of p-PI3K, p-AKT and p-mTOR. (NS indicates a significant difference at *p* > 0.05, and * indicates a significant difference at *p* < 0.05, *n* = 3).

**Figure 7 toxins-15-00427-f007:**
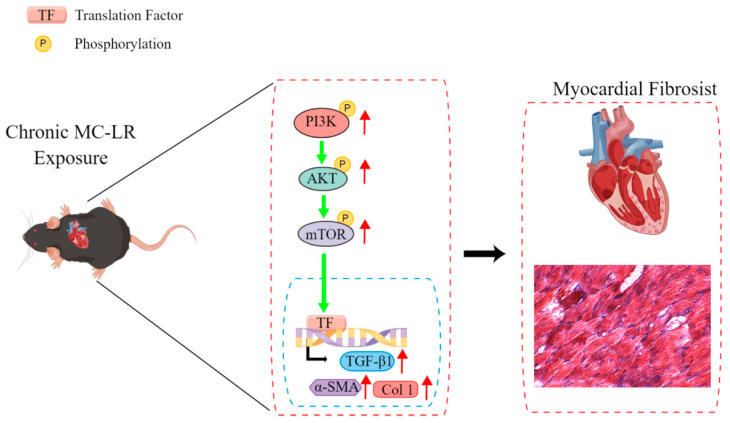
Long-term environmental levels of MC-LR exposure induced myocardial fibrosis through the PI3K/AKT/mTOR signaling pathway (red arrows indicate the elevated expression, and green arrows indicate the activation).

**Table 1 toxins-15-00427-t001:** Echocardiography results (* indicates a significant difference at *p* < 0.05, ** indicates a significant difference at *p* < 0.01, and *** indicates a significant difference at *p* < 0.001; *n* = 3).

Groups	Control (µg/L)	MC-LR Treated (120 µg/L)
LVEF (%)	53.88 ± 3.96	36.28 ± 4.32 ***
LVFS (%)	27.36 ± 2.38	17.24 ± 2.39 ***
LVESD(mm)	29.4 ± 7.55	49.58 ± 2.90 **
LVEDD (mm)	63.02 ± 12.22	77.95 ± 4.43 *
LVPWd (mm)	0.57 ± 0.10	0.72 ± 0.07
LVPWs (mm)	0.88 ± 0.17	0.89 ± 0.11
IDd (mm)	3.81 ± 0.33	4.18 ± 0.10
LVIDs (mm)	2.78 ± 0.32	3.46 ± 0.08
HR (beat/min)	403.28 ± 43.69	388.00 ± 26.03

**Table 2 toxins-15-00427-t002:** Primer sequence required for the experiment.

Primer Name	Forward Primer	Reverse Primer
β-actin	CTAAGGCCAACCGTGAAAAG	AGGGTCCCAGACAGAAGTTG
Col1	TGTTCAGCTTTGTGGACCTC	TCAAGCATACCTCGGGTTTC
α-SMA	CCTGAAGAGCATCCGACACT	TACGTCCAGAGGCATAGAGG
TGF-β	ACCGGAGAGCCCTGGATA	CCACGTAGTAGACGATGG

**Table 3 toxins-15-00427-t003:** Antibodies and sources required for experiments.

Antibody Name	Source	Dilution Ratios
TGF-β	Proteintech, Wuhan, China	1:1000
α-SMA	Proteintech, Wuhan, China	1:1000
COL1	Proteintech, Wuhan, China	1:1000
PI3K	Proteintech, Wuhan, China	1:1000
P-PI3K	Cell Signaling Technology Company, Danvers, MA, USA	1:1000
AKT	Proteintech, Wuhan, China	1:1000
P-AKT	Cell Signaling Technology Company, Danvers, MA, USA	1:1000
mTOR	Proteintech, Wuhan, China	1:1000
p-mTOR	Proteintech, Wuhan, China	1:1000
β-ACTIN	Cell Signaling Technology Company, Danvers, MA, USA	1:1000
Rabbit secondary antibody	Abbkine company, Wuhan, China	1:3000
MC-LR	Alexis Corporation (Lausen, Switzerland)	1:3000
MMP9	Proteintech, Wuhan, China	1:1500

## Data Availability

The data presented in this study are available in this article.

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
