# Peer review of "Cardiac Toxicity Induced by Long-Term Environmental Levels of MC-LR Exposure in Mice"

_toxins, 2023, doi:10.3390/toxins15070427_

Round 1
Reviewer 1 Report
In this article, the authors have investigated the toxic long term, low dose exposure effects and mechanisms of MC-LR on murine heart and the possible role in cardiovascular disease. Following are my comments on the paper:
Major comments:
Section 2.1.1: Authors mention that MC-LR was provided in drinking water. Microcystin is known to adsorp to plastic than glass and also degrades to some extent at room temperature. Please mention how often was the water changed or spiked with MC-LR to maintain the required concentration?
Authors have mentioned in detail the techniques for qPCR, histology and Western Blot. Please mention what specific area of the heart was chosen for each procedure.
Section 3.1: Authors show data for the detection of MCLR using WB. Please specify the antibody used for this in the Materials and Methods section (2.2.5)
The daily water consumption of mice is 4 ml. Please verify the values mentioned in Line 228. Seems a little excess for 1 mouse.
Please correct Figure 2B. Related text mentions about a graph showing consumption of drinking water volume.
Figure 4: Please clarify if the heart weights for each mouse are normalized to their body weight.
Figure 5: Please point out the H&E (A and B) vs Masson (C and D) stained panels in the figure and legend.
It would be good to see if the expression of MMP9, late fibrosis marker, also goes up after a chronic exposure to the toxin.
Minor Comments:
Line 135- Correct to H&E
Check spacing between words throughout
Please mention the n in the figure legends.
Line 254: Start the sentence with Exposure of mice to MC-LR... instead of infection of mice...
Line 316-317: Please correct the statement structure.
Line 321: trahira is the common name of Hoplias malabaricus. Please correct to Trahira (Hoplias malabaricus).
Also correct Lines 354 and 377 for spacing and sentence structure.
Reviewer 2 Report
The authors rise up an interesting side related to effects of MC-LR (cyanotoxins) on cardiovascular health. They based their study on a long term exposure to microcystins to evaluate the impact and the toxic mechanisms of MC-LR on heart’s health.
The study pointed out an up regulation of both markers (COL-1, TGF-β1, α-SMA) and signaling pathway molecules PI3K/AKT/mTOR indicating myocardial fibrosis.
Recommendation to be considered in order to improve the manuscript:
- Line 32 correct Oscillatoria instead of Oscillator
- Revise the figures titles to be more explicit (Figure 1 to 4)
- Figures 6, 7 and 8: change “(*, P<0.05, n=3)” by: * indicate significant difference at P<0.05, n=3.
- Figure 9: give adequate legend (differences between red/ green lines; P significance). The figure is suitable to be cited as graphical abstract instead of being part of conclusion.
- Revision of references list (citation of references according to Authors guidelines
Reviewer 3 Report
The manuscript presents a novel and needed toxicology study using a mouse-based animal model to explore the relationship between consumption of the cyanotoxin MC-LR and indicators of cardiotoxicity. The manuscript and data included within address a gap in the current literature where there is scant information available regarding MC-LR and cardiotoxicity. The evidence presented was done in an experimental design that enabled comparison against a gradient and with a negative control and high exposure group as well as concentrations that are observable in nature. The study speaks specifically to mice and leaves it to others to speculate what such results could mean for human exposure/human consumption of cyanotoxins, notably MC-LR. Overall, while clear dose-response relationships were not consistently available for the many markers or indicators of potential cardiotoxicity, differences were observable between the negative control and even low concentration treatment groups, and very apparent in the high treatment group. The histological imagery illustrates clearly that these toxins, especially at high concentrations used in this study, can illicit adverse effects, and less apparent histological effects on the heart are plausible, and function was impaired. These results are a valuable contribution; however, there were several major issues, moderate issues, and minor/grammatical issues that presumably should be addressed to make this a more impactful and well-described study for others to use. These issues are described below.
Major Recommendations:
(1) In the introduction, an important reference is missing. The animal ethics committee approval is dated for 2018, so it is possible that the background research used in writing this manuscript is a little dated. One statement needing citation or further consideration is from the manuscript introduction: “To date, information regarding the cardiotoxic effects of MC-LR, especially low concentration of long term of MC-LR exposure, remains scanty and the underlying mechanism is still unclear.” – The authors of the below-mentioned mini-review indicate that the literature is “scant” on this subject. The authors are encouraged to review this publication. Alosman, M., Cao, L., Massey, I.Y. and Yang, F., 2021. The lethal effects and determinants of microcystin-LR on heart: A mini review. Toxin reviews, 40(4), pp.517-526.
(2) Methods: Water Prep: A key part of the study focuses on the dose-response relationship, and it is important to provide more detail on the dose aspect, specifically, the water prep. Details on the preparation of the toxin-spiked water would be of value to readers and standardization of a protocol for future study. The concentrations selected by the researchers are appropriate for comparison to WHO standard and various state and national standards; however, more information is needed in regard to (A) the stock source of the MC-LR, (B) how the MC-LR was mixed into the water, (C) how often the preparation was made during the study or if a single preparation was used for each concentration and dispensed over the study. (D) Additionally, if there were any QA/QC protocols to ensure the MC-LR concentrations were 0, 1, 30 ug/L, etc., and if so, (E) how were concentrations measured in water after preparation to verify concentration?
(3) Western Blot details: In the results, Figure 1, the Western blot shows MC-LR detection. It is not clear in the methods section, which antibody and/or antibodies were used for detecting/observing MC-LR via Western blot analysis. Additionally, the text should reference Table 2 near line 201-203 rather than the Table not being connected by a reference in text.
(4) Results: In regard to cardiac function, section 3.5., the authors report a significant difference between the highest concentration treatment group; however, there is no mention if there were any other differences or the absence of differences in the other treatment groups. Clarifying no other significant differences in the lower concentration treatment groups, if that is the case, would be helpful for properly understanding the results of the study as an outside reader.
(5) Similar to the prior concern, section 3.6., the authors report observed adverse cardiac histology in the mice with the MC-LR spiked water; however, the authors do not provide details on whether or not this observed characteristic was apparent only in the highest concentration. The figure (Figure 5) presents only the control and highest treatment concentration conditions. Clarifying the presence or absence of appreciable changes from the control condition relative to the lower MC-LR treatment groups would be helpful.
Moderate Concerns/Recommendations:
(1) Figure 6 and 7 captions: Indicates p<0.05, should this indicate relative to the control group, it does so in text, but maybe helpful in Figure note as well. Figure 7 caption: definition is needed for the NS (not significant) and the double asterisk **.
(2) Line 321: need to specify trahira are a type of fish. Also, scientific nomenclature needs italicized. “into the fish species trahira (Hoplias malabaricus).” Currently it says trahira and H. malabaricus, however, these are the same, aren’t they?
(3) Line 324: Need to provide again the name of the fish, and specific epithet of scientific nomenclature needs lowercase, and Oryzias latipes needs italicized. …”the heart rate of the Japanese rice fish (Oryzias latipes) <- italicize.
(4) Line 325: Need to provide latin for Zebrafish. ‘4.0 mM MC-LR, zebrafish (Danio rerio) larvae…”
(5) I’m unsure about this recommendation, but could there or should there be communication about the generalizability of the results to other mice, animals, including humans, and that further research would be warranted; however, a precautionary approach is suggestive that cardiotoxicity is plausible in other animals from consuming MC-LR.
Minor/Grammar:
(1) Section 3.6. Line 254: Is “infection” of the mice the appropriate term. MC-LR is a chemical rather than a pathogen, so wouldn’t “Dosing of mice” or “Exposure of mice” be more accurate.
(2) Table 3: It is a little difficult to read horizontally. If the authors invert the groups and ECG tests and have a more vertical table than horizontal, it would likely improve the illustration of the results.
(3) Abstract: Line 5: remove ‘as’. Should read: “…are considered a serious global problem”
(4) Introduction: Line 49: Should read as “Previous research has shown…”
(5) Introduction: Line 54: replace induce with… “induces” cardiovascular toxicity
(6) Results: Line 270: should Col1 read as “COL1” mRNA
(7) Discussion: Line 316: Needs reworded. “could decrease ‘heart function in mice’ and this study ‘is likely the first or among the first to demonstrate such decrease….???”
(8) Conclusion section: missing period in very last statement of manuscript.
Reviewer 4 Report
The review is attached.

Round 2
Reviewer 3 Report
Overall, all the edits and/or revisions requested appear to have been made and of those made, they have been done satisfactorily from my point of view.